# Influence of Nanoparticles on Thermophysical Properties of Hybrid Nanofluids of Different Volume Fractions

**DOI:** 10.3390/nano12152570

**Published:** 2022-07-27

**Authors:** Mohd Zulkifly Abdullah, Kok Hwa Yu, Hao Yuan Loh, Roslan Kamarudin, Prem Gunnasegaran, Abdusalam Alkhwaji

**Affiliations:** 1School of Mechanical Engineering, Universiti Sains Malaysia, Engineering Campus, Nibong Tebal 14300, Malaysia; yukokhua@usm.my (K.H.Y.); lohhaoyuan@student.usm.my (H.Y.L.); roslan_k@usm.my (R.K.); 2Institute of Power Engineering, Putrajaya Campus, Universiti Tenaga Nasional, Jalan IKRAM-UNITEN, Kajang 43000, Malaysia; Prem@uniten.edu.my; 3School of Mechanical and Industrial Engineering, Alasmarya Islamic University, Zliten 218521, Libya; alkhwaji@asmarya.edu.ly

**Keywords:** nanofluid, density, thermal conductivity, viscosity, specific heat capacity

## Abstract

Nanofluids are frequently employed in numerous heat transfer applications due to their improved thermophysical properties compared to a base fluid. By selecting suitable combinations of nanoparticles, hybrid nanofluids can have better thermophysical properties than mono nanofluids. Thus, this study examines the effect of volume fractions of hybrid nanofluids on different thermophysical properties, such as density, thermal conductivity, specific heat, and dynamic viscosity. Thermophysical properties of copper–nickel (Cu–Ni) water-based hybrid nanofluids are determined using molecular dynamic (MD) simulation for different volume fractions of 0.1–0.3%. Results show that the density, thermal conductivity, and viscosity of Cu–Ni hybrid nanofluids increase with volume fraction, whereas the specific heat capacity at a constant pressure decreases with volume fraction. These properties are validated for the base fluid, mono nanofluids, and hybrid nanofluids. Results are in good agreement with previous findings. The thermophysical properties of Cu–Ni hybrid nanofluids significantly improve and have better characteristics for cooling fluids than the base fluid.

## 1. Introduction

Nanofluids consist of nano-sized particles suspended in a base fluid, such as water, engine oil, and organic fluids, that are commonly used as working fluids in heat transfer applications. Nanoparticles have a large surface-area-to-volume ratio, small size, and great stability when suspended in a base fluid. Thus, the diffusion of nanoparticles in nanofluids causes enhanced thermophysical properties [1]. Choi first studied the enhancement of thermal conductivity of nanofluids, which are then listed as heat transfer fluids, bio- and pharmaceutical nanofluids, or environmental and medicinal nanofluids. They discovered that a fluid’s thermal conductivity may be doubled or tripled [2]. At present, nanofluids are used in certain heat transfer applications such as plate heat exchangers, coolant materials [3], and tribology [4]. By comparison, hybrid nanofluids are formed by suspending two different nanoparticles in the base fluid. Selecting the proper combination of nanoparticles can utilize the synergistic effect and thus obtain considerable benefits, such as having exceptionally high thermal conductivity [5]. The heat transfer performance of any fluid depends on its thermophysical properties. The volume fraction of mono- and hybrid nanofluids have varying effects in different thermophysical properties. Recently, Wilk et al. showed that the density and viscosity of copper–water (Cu–H_2_O) nanofluids increase with volume fraction at the same temperature [6]. Mostafizur et al. experimentally investigated methanol-based TiO_2_ nanofluid and found that its density increases with volume fraction [7]. Fadhilah et al. found that the thermal conductivity of Cu–H_2_O nanofluid in the automotive cooling system increases with volume fraction [8]. Sundar et al. conducted an experimental investigation on the thermal conductivity and viscosity of nickel (Ni) nanofluid and found that both properties increase with volume fraction [9]. Rajabpour et al. observed that the specific heat of Cu–H_2_O nanofluid decreases with volume fraction [10]. In addition, Shahrul et al. reviewed experimental and theoretical studies on specific heat of nanofluids [11] and the results show contradictory outcomes, with the specific heat of most of nanofluids decreasing with volume fraction. On the other hand, Pak and Cho [12] showed reliable results of the density of mono nanofluids with their experimental model [13,14]. However, it was discovered that the dispersed fluid’s convective heat transfer coefficient at a volume concentration of 3% was 13% lower than that of pure water. Takabi et al. then extended the mixture rule and suggested a model to determine the density for hybrid nanofluids [15]. Their findings concur with those of Sundar et al. [16,17] in that hybrid nanofluid densities increase with volume fraction and fall between those of the corresponding mono nanofluids. However, the greatest error for the Nusselt number correlations for nanofluid and hybrid nanofluid was around 11% and 12%, respectively. Furthermore, Sarkar et al. reviewed the thermal conductivities of hybrid nanofluids [18], most of which increased with volume fraction and were higher than those of mono nanofluids and the base fluid. Gao et al. experimentally investigated aqueous solutions of graphene oxide with alumina (GO-Al_2_O_3_) hybrid nanofluids using a cooling method [19] and found that their specific heat decreased with volume fraction and was higher than that of mono nanofluids. Suresh et al. showed that the viscosity of alumina (Al)–Cu–H_2_O hybrid nanofluid increased with volume fraction and was higher than that of Al–H_2_O mono nanofluid. They discovered that when 0.1% Al_2_O_3_-Cu/water hybrid nanofluids were compared to water, the average increase in friction factor was 16.97% [20]. Moreover, Babar et al. reviewed the viscosity of hybrid nanofluids, several of which showed higher viscosity than that of mono nanofluids, whereas others showed viscosity between those of their respective mono nanofluids [21]. These studies show that the density, viscosity, and thermal conductivity of most of the mono and hybrid nanofluids increased with volume fraction, whereas the specific heat decreased with volume fraction. Wang and Chen [22] also utilized Al_2_O_3_-water nanofluid to numerically study heat transport. They investigated the effect of different nanofluid volume fractions and geometric parameters on the inlet and outlet pressure drop, flow resistance coefficient, substrate temperature, Nusselt number (*Nu*), and system thermal resistance in the fractal microchannel. The result showed the Al_2_O_3_ nanofluid with a volume fraction of 5% had a 12.5–14.7% lower thermal resistance than deionized water.

However, the experimental work to determine the thermophysical properties of nanofluids is extremely time-consuming and costly due to numerous and complicated correlations between the properties of particles. In addition, the interpretation of experimental data is very difficult, especially for a complex system. In this study, the thermal conductivity, specific heat, dynamic viscosity, and density of Cu, Ni, and Cu–Ni nanofluids are determined at different volume fractions by molecular dynamic (MD) simulation using a large-scale atomic/molecular massively parallel simulator (LAMMPS). As a result, the work makes it easier and faster to anticipate the thermophysical characteristics of mono and hybrid nanofluids. Additionally, it is possible to investigate the physical understanding of how particles interact with the base fluid molecules.

## 2. Dynamic Viscosity

MD simulation numerically obtains the atomic positions by solving the differential equation of the second law of Newton and inter-atomic potential forces. However, the Green–Kubo (GK) method is used to determine the properties of fluids in MD simulation using LAMMPS. In this study, the GK method with the canonical constant of the atomic number, pressure, and temperature (NPT) ensemble is used to determine the thermal conductivity, specific heat, and dynamic viscosity of the base fluid, which is de-ionized water for validation. The NPT ensemble is used for its resemblance to laboratory conditions, which are constant pressure and temperature throughout the time step. Adding nanoparticles into base fluids affects the overall properties. Thus, in this study, relations between properties are investigated. Thermal conductivity and specific heat are related to the total system energy, while dynamic viscosity is related to a stress tensor of particles, which is also found in the GK method. The GK relation for viscosity simulation is defined by [23]:(1)$$\mu_{xy}=\frac{V}{k_{B}T}\int_{0}^{\infty }\langle P_{xy}\left(0\right)P_{xy}\left(t\right)\rangle dt$$
where μ is the viscosity, *V* is the volume of system, *k_B_* is the Boltzmann constant that averages over the ensemble, and *P* is the off-diagonal element of stress tensor.

In this study, the methodology is separated into two parts: MD simulation and data analysis. MD simulation is used to obtain the density, total energy, and dynamic viscosity of the system. The simulation starts with the base fluid because its actual properties are known and are used to improve the reliability of results. The relations between the simulated total energy and the actual thermal conductivity, simulated changes in the total energy and the actual specific heat, and the simulated and actual dynamic viscosities of water are obtained. Then, the total energy and viscosity of mono and hybrid nanofluids are simulated and the thermophysical properties are obtained using the relations shown in the base fluid. For the density, the error between the simulation and actual results is very small and thus considered to have no relation. Details are discussed in the following sections.

### 2.1. Thermal Conductivity

The GK relation is also used to obtain the thermal conductivity property. The GK relates the auto correlation function (ACF) to transport coefficients during the equilibrium state [15]. The GK method is an empirical mode decomposition (EMD) approach for obtaining thermal conductivity from the relationship between heat flux and equilibrium state autocorrelation function [23,24,25]:(2)$$k=\frac{1}{3Vk_{B}T^{2}}\int_{0}^{\infty }\langle J_{i}\left(0\right)J_{i}\left(t\right)\rangle dt$$
where *k*, *V*, *T*, and *k_B_* are the thermal conductivity, volume of the simulation box, system kinetic temperature, and Boltzmann constant, respectively. The heat flux, *J*, refers to the per- kinetic, potential, and virial atom contributions from non-bond, bond, angle interaction, and auto correlation functions. The heat flux, *J*, is defined below: (3)$$J=\left[\overset{N}{\underset{j=1}{\sum }}v_{j}E_{j}-\overset{N}{\underset{a=1}{\sum }}h_{a}\overset{N_{a}}{\underset{j=1}{\sum }}v_{aj}\right]+\frac{1}{2}\left[\overset{N}{\underset{i=1}{\sum }}\overset{N_{a}}{\underset{j=1,j\neq i}{\sum }}r_{ij}\left(v_{j}\cdot F_{j}\right)\right]$$
where *vj* is the velocity of particle *j*, *Ej* is per atom energy for kinetic and potential, *h_a_* is the average partial enthalpy of species *α*, *rij* and *Fij* are the displacement and interacting forces between particles *i* and *j*, respectively, and *N* is the total number of particles [23,24,25]. The mean partial enthalpy refers to the total amount of kinetic energy, potential energy, and virial total per particle. In Equation (6), the kinetic and potential terms of the heat flux convey the transported energy, while the virial contribution reflects work carried out using the stress tensor [23,24,25] below: (4)$$h_{a}=\frac{1}{N_{a}}\overset{N_{a}}{\underset{j=1}{\sum }}\left(E_{j}+r_{j}\cdot F_{j}\right)$$

### 2.2. Isobaric Specific Heat Capacity

Figure 1 illustrates the simulated nanofluid domain temperature (K) and total energy (kcal/mol) for two different conditions under which the system was initially at 283 K and then increased to 293 K at a constant 1 atm pressure.

During the NPT ensemble equilibrium, the isobaric thermal capacitance is calculated for the simulated H_2_O domain to increase the H_2_O molecule temperature from T1 to T2 and the associated total energy *E1* and *E2*, respectively [23,24].
(5)$$C_{P}=\frac{E_{2}^{tot}-E_{1}^{tot}}{T_{2}-T_{1}}+\frac{\Delta Q}{\Delta T}$$
where *Etot* is the total energy per molecule and Δ*Q*/Δ*T* is the quantum contribution of intramolecular vibrational moles to the isobaric specific heat capacity. 

### 2.3. Atomic and Molecular Set-Up

Moltemplate software is used to generate the water molecule, known as the simple point charge extended (SPCE) of water model [26], which specifies a three-site rigid water molecule. Thus, there are three interaction points. Charges and Lennard–Jones (LJ) parameters are assigned to each of three atoms in the molecule. ‘Fix rattle’ command is then used to hold the oxygen–hydrogen (O–H) bonds at rigid angles such that the molecules would not break and are not flexible. The simulation box generated is 34.5 Angstrom on each side. The system contains 1000 water molecules or 3000 atoms with a well-dispersed state. In addition, a periodic boundary condition (PBC) is applied in three dimensions of the system to avoid the problem of bonds crossing the simulation box boundary. Applying PBC can simulate an infinite system at which real life conditions can be resembled. The number of atoms required for different volume fractions of mono and hybrid nanofluids is determined as 0.1%, 0.2%, and 0.3%. In LAMMPS, the total energy is affected by the number of atoms in the system. Given that relations are used to obtain the results of thermophysical properties, the number of atoms for the total energy simulation system is 3000 atoms. 

The atoms are placed at certain distances in the system for dispersion. Figure 1 shows a 0.3% volume fraction of a Cu nanofluid system as an illustration using OVITO software. The molar masses for hydrogen, oxygen, copper, and nickel particles are also defined in this step. Table 1 shows the molar masses for the four different atoms.

Next, interatomic potentials are defined for the interaction among atoms. The LJ potential is used to describe the interactions of H–H and O–O atoms. The LJ potential is defined by [27]:(6)$$U\left(r_{ij}\right)=4\epsilon \left[{\left(\frac{\sigma }{r_{ij}}\right)}^{12}-{\left(\frac{\sigma }{r_{ij}}\right)}^{6}\right]\left(r_{ij}<r_{cutoff}\right)$$
where *r_ij_* is the distance between atoms *i* and j, and ϵ and σ are LJ potential parameters, representing the interaction strength and interatomic length scale, respectively. In this calculation, the first term is responsible for repulsion at a short distance and the second term is responsible for attraction at long distances. Coulombic interaction is also involved with the LJ potential to describe the interactions of H–H and O–O atoms. For cross interaction between the H and O atoms, the Lorentz–Berthelot (LB) mixing rule is used and is defined by [28]:(7)$$\epsilon_{ij}=\sqrt{\epsilon_{i}\epsilon_{j}}$$
(8)$$\sigma_{ij}=\frac{1}{2}\left(\sigma_{i}+\sigma_{j}\right)$$

Table 2 shows the LJ potential parameters used for H–H, O–O, and H–O interactions.

Kang et al. used the embedded atom model (EAM) potential for Cu–Cu interaction and, because the metallic bonding is considered, it posits greater accuracy than the LJ potential [29]. Thus, this study uses the EAM potential for interaction of Cu–Cu and Ni–Ni atoms. EAM potential can be used to compute the pairwise interactions of metallic atoms and is defined by [29]:(9)$$E_{i}=F_{\alpha }\left(\overset{\;}{\underset{j\neq i}{\sum }}\rho_{\beta }\left(r_{ij}\right)\right)+\frac{1}{2}\overset{\;}{\underset{j\neq i}{\sum }}\emptyset_{\alpha \beta }\left(r_{ij}\right)$$
where *F* is the embedding energy, which is a function of the atomic electron density ρ, ∅ is the pair potential interaction, and *α* and *β* are the element types of atoms *i* and *j*. The use of EAM potential is assumed to prevent the agglomeration of Cu and Ni atoms during simulation because the van der Waals forces between atoms is not calculated. For EAM potential parameters, the ‘pair_coeff’ command is used with the input of selected EAM potential files to enable reading and parsing in LAMMPS. Table 2 shows the EAM potential files used for Cu–Cu and Ni–Ni interactions. 

For cross interaction between Cu–H_2_O, Ni and H_2_O, and Cu–Ni, the LB mixing rule cannot be used because of the involvement of the LJ and EAM potentials. Thus, the LJ potential with cut-off is used for those interactions. The parameters for those interactions have not been found in previous research, and are thus logically estimated in this simulation. Further study is needed to obtain the accurate parameters. Table 3 shows the LJ potential parameters used for Cu–H_2_O, Ni–H_2_O, and Cu–Ni cross-interactions.

Then, random initial velocities are assigned to the atoms in the system before the simulation. The velocity Verlet algorithm, which requires a less computational amount for simulation, is also used as the numerical method to solve the differential equations of motion [28]. Then, the NPT ensemble is used such that the number of particles, pressure, and temperature are kept constant throughout the simulation. For the total energy system simulation, the time step is set at 500,000 ps and its size is set at 2 ps. The time step is sufficient for this simulation because results from total energy simulation converge in the time set-up. 

## 3. Results and Discussion

In this study, the NPT ensemble was applied and determined that the temperature was not constant but rather varied in a certain range. The results obtained from MD simulation were also plotted using MATLAB for every time step. For the density and total energy of the system, the average values throughout the time step were taken as the predicted results. Figure 2, Figure 3 and Figure 4 show the temperature, density, and total energy of system over the time step, respectively. For dynamic viscosity, the results converged slowly because of the complex calculation in GK relation. The results that were nearly constant throughout the time step were considered and averaged. 

### 3.1. Effect of Volume Fraction in Thermal Conductivity

Figure 5 shows the thermal conductivity versus volume fractions for Cu, Ni, and Cu–Ni nanofluids. The results show that the thermal conductivity of Cu and Ni nanofluids increased with the volume fraction, which is consistent with the literature [8,9]. The thermal conductivity of mono nanofluids was also larger than that of the base fluid, except for the 0.1% volume fraction of Ni nanofluid. The figure shows that the linear curve slightly differs from the actual curve, thus causing the slightly lower result of the 0.1% volume fraction Ni nanofluid. Overall, the thermal conductivity of mono nanofluids was larger than that of the base fluid, which is consistent with the literature [30]. For the Cu–Ni hybrid nanofluid, thermal conductivity increased with volume fraction and was larger than that of the base fluid and their respective mono nanofluids. This result is consistent with the recent literature review by Sarkar et al. [18]. 

The increase in thermal conductivity of nanofluids was mainly due to the Brownian motion of nanoparticles, which caused the particles to absorb heat from the surrounding base fluid and thus enhanced the thermal transport of nanofluids. In addition, due to the synergistic effect of hybrid nanoparticles, suspending dissimilar nanoparticles in the base fluid resulted in greater thermal conductivity than the enhancement caused by suspending only one type of nanoparticles.

### 3.2. Effect of Volume Fraction in Specific Heat

A change in the total energy is related to the product of specific heat and temperature difference. The specific heat values of Cu and Ni nanofluids decreased with volume fraction and were lower than those of the base fluid (Figure 6). These results are consistent with the literature [10,11]. For the Cu–Ni hybrid nanofluid, specific heat decreased with the volume fraction. The specific heat of Cu–Ni hybrid nanofluids at any volume fraction was also lower than that of the base fluid but was higher than that of Cu and Ni nanofluids for the same volume fraction. These results are consistent with those obtained from GO–Al_2_O_3_/H_2_O hybrid nanofluids by Gao et al. [19]. 

The specific heat values of Cu and Ni were 385 J/kg.K and 440 J/kg.K, respectively, which were much lower compared with H_2_O, which was approximately 4184 J/kg.K at ambient temperature. When nanoparticles were suspended in the base fluid, the heat was absorbed by these low specific heat nanoparticles and thus reduced the specific heat of the nanofluids. When the volume fraction increased, the suspended nanoparticles increased and a greater amount of heat was absorbed, causing lower specific heat of the nanofluid. The specific heat of Cu was lower than Ni. Thus, the specific heat of Cu nanofluid was lower than that of Ni nanofluid, as illustrated in Figure 6. For hybrid nanofluids, the specific heat was larger than that of mono nanofluids possibly due to the synergistic effect between Cu and Ni nanoparticles, which caused a significant increase in the overall specific heat.

### 3.3. Effect of Volume Fraction in Dynamic Viscosity

The GK method is also used with NPT ensemble to predict dynamic viscosity of the base fluid compared with its actual viscosity [31]. Figure 7 shows the comparison between simulated and actual dynamic viscosity of the base fluid at different temperatures. The results show that the predictions differed from actual results. The figure shows that simulated viscosity of base fluid had good agreement with the published viscosity [31] with approximately 96%. Viscosity of mono and hybrid nanofluids was simulated at a temperature of 20 °C. Figure 8 illustrates that the actual viscosity against volume fraction for Cu, Ni, and Cu-Ni nanofluids increased almost linearly with volume fraction.

The results show that the viscosity of nanofluid increased with volume fraction. The smaller case was that the linear relation differed slightly from the actual curve, because the simulated viscosity of 0.1% volume fraction of Cu nanofluid was slightly larger than that of H_2_O (the base fluid). Overall, the result is consistent with those obtained by Wilk et al. [6]. Further research is needed to determine more suitable parameters. For the Cu–Ni hybrid nanofluid curve, the viscosity increased with volume fraction and was larger than that of its individual nanofluids and the base fluid. The result is consistent with those obtained by Suresh et al. [20].

Overall, viscosity of hybrid nanofluids increases with volume fraction. It is suspected that when volume fraction increases, more nanoparticles are involved and thus more random Brownian motions of nanoparticles disrupt the moving of fluid particles, with a tendency to increase the viscosity of the entire system. 

### 3.4. Effect of Volume Fraction on Density

The density of the base fluid at temperature of 20 °C was simulated and compared with its actual value for validation. Table 4 shows that the simulated density of water at 20 °C was very close to the actual density. The error was calculated using Equation (10), below:(10)$$Error=\frac{\left|Simulated\;value-Actual\;value\right|}{Actual\;value}$$

The error obtained was 0.1643%, which shows that the density calculation by MD simulation was accurate and reliable. Thus, no relation or calibration was needed for density simulation. The densities of nanofluids were simulated at a temperature of 20 °C for comparison. Figure 9 shows the density against volume fraction of Cu, Ni, and Cu–Ni nanofluids.

The result shows that the density of the nanofluid increased with volume fraction and was larger than that of the base fluid. This result is also consistent with the literature [6,7]. Ni nanofluid showed the same result as Cu nanofluid. From the Pak and Cho model [12], the densities of Ni/water nanofluid at 0.1% and 0.3% volume fractions were 1.0064 g/cm^3^ and 1.0222 g/cm^3^, respectively. By comparing the current results with the literature, the difference obtained was 0.0298% and 0.4109% for 0.1% and 0.3% volume fractions, respectively, which shows that result of density obtained using MD simulation was reliable and accurate. The densities of nanoparticles were larger than those of the base fluid. Thus, nanofluids could achieve higher density by dispersing high density nanoparticles into the base fluid.

For the Cu–Ni hybrid nanofluid, the density increased with volume fraction and was larger than that of the base fluid. In addition, for the same volume fraction, the density of the Cu–Ni nanofluid was higher than that of the Ni nanofluid but lower than that of the Cu nanofluid. This was because the density of Cu nanoparticles was the largest, followed by the combination of Cu–Ni nanoparticles, and then Ni nanoparticles. This result is consistent with those of Takabi et al. [15], which predicts that the density of an hybrid nanofluid is higher than that of the base fluid but lower than that of the Cu nanofluid.

## 4. Conclusions

The present study aimed to determine the thermophysical properties, i.e., the density, thermal conductivity, specific heat, and dynamic viscosity, of Cu, Ni, and Cu–Ni water-based nanofluids at different volume fractions of 0.1–0.3% using MD simulation. The findings are as follows.

➢The thermal conductivity of the Cu–Ni hybrid nanofluid increased with volume fraction and was larger than that of the base fluid and both mono nanofluids.➢The specific heat of the Cu–Ni hybrid nanofluid decreased with volume fraction and was lower than that of the base fluid but higher than that of both mono nanofluids.➢The viscosity of the Cu–Ni hybrid nanofluid increased with volume fraction and was higher than that of the base fluid and both mono nanofluids.➢The density of the Cu–Ni hybrid nanofluid increased with volume fraction and was higher than that of the base fluid and the Ni nanofluid but lower than that of the Cu nanofluid.

Overall, the obtained density, thermal conductivity, specific heat, and viscosity of hybrid nanofluids for different volume fractions are consistent with the published literature. The volume fraction of hybrid nanofluid had a significant influence on the thermopysical properties and could affect the heat transfer performance, such as the Nusselt number, friction factor, and pressure drop.

## Figures and Tables

**Figure 1 nanomaterials-12-02570-f001:**
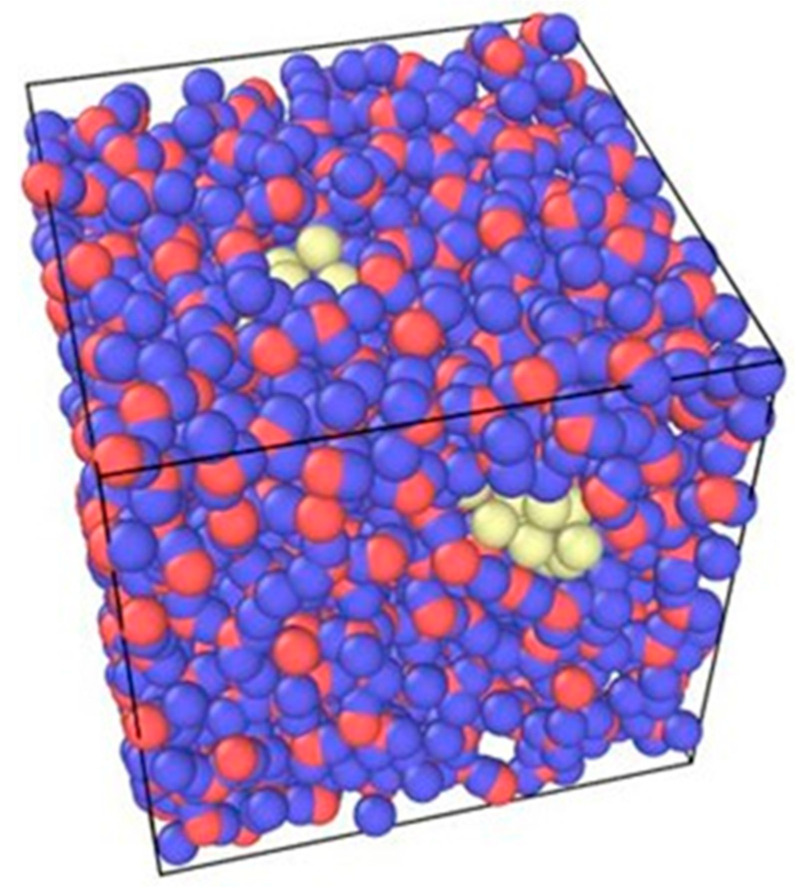
Illustration of water and copper particles.

**Figure 2 nanomaterials-12-02570-f002:**
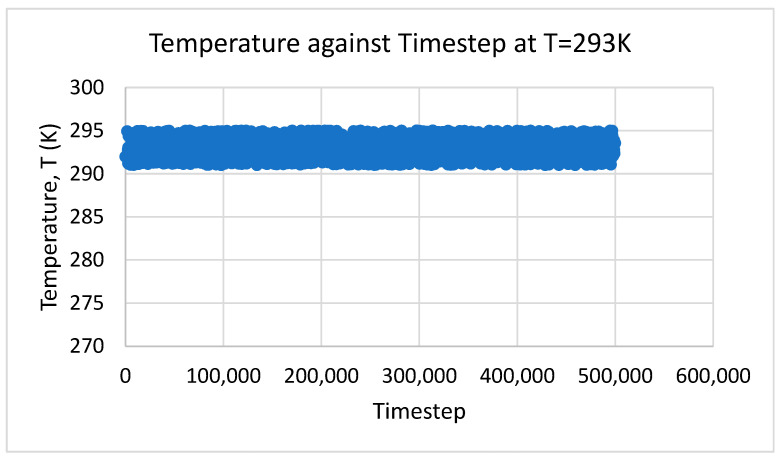
Temperature against time step.

**Figure 3 nanomaterials-12-02570-f003:**
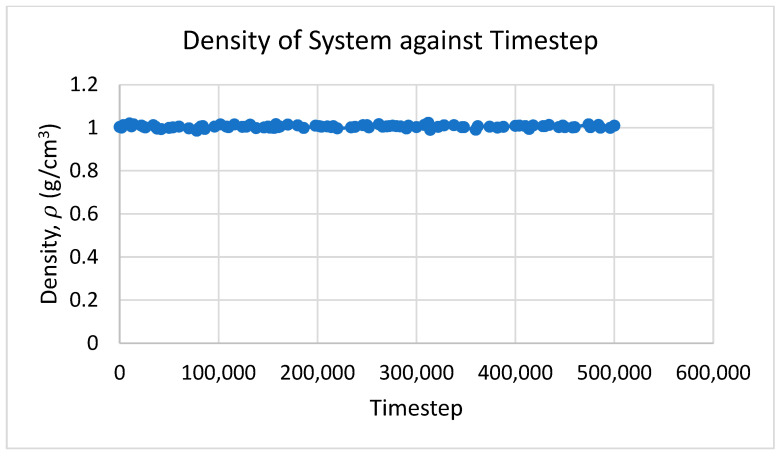
Density of system against time step.

**Figure 4 nanomaterials-12-02570-f004:**
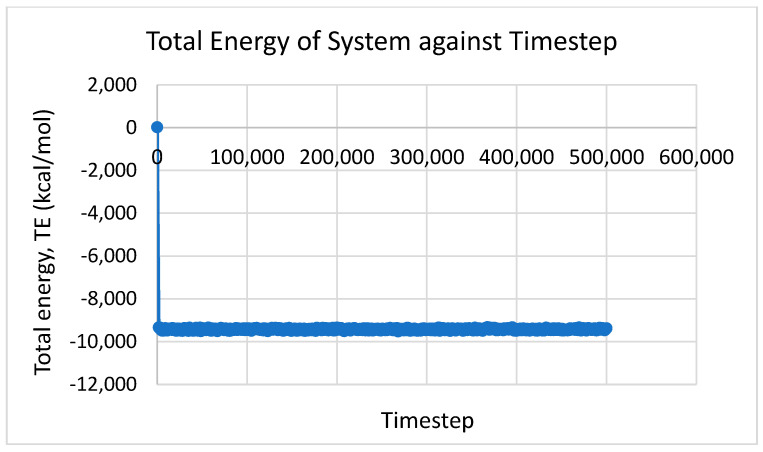
Total energy of system against time step.

**Figure 5 nanomaterials-12-02570-f005:**
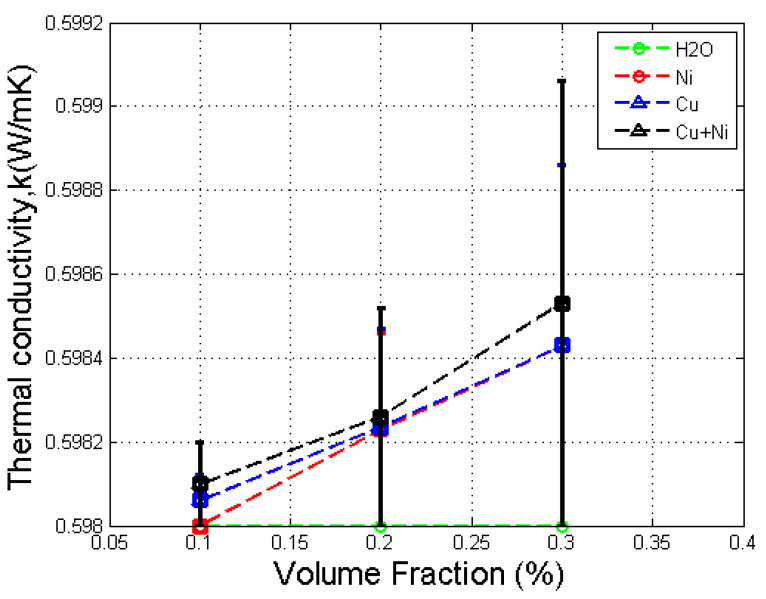
Graph of thermal conductivity against volume fraction for Cu, Ni, and Cu-Ni nanofluids with error bars.

**Figure 6 nanomaterials-12-02570-f006:**
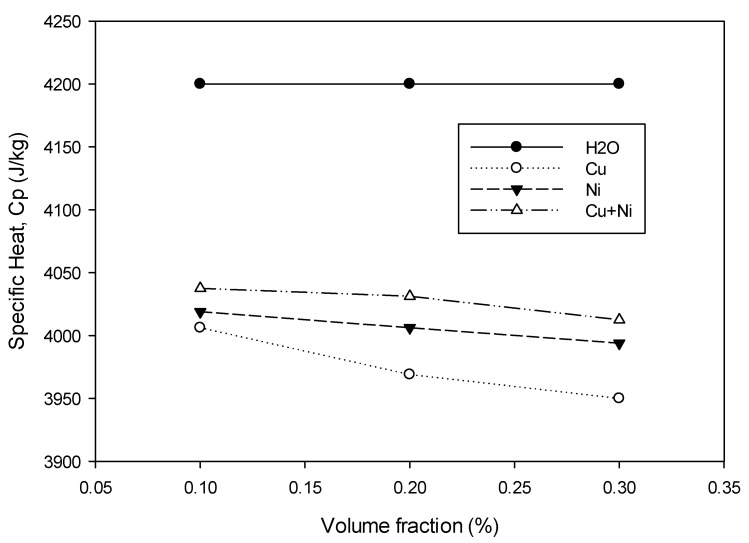
Graph of specific heat against volume fraction for Cu, Ni and Cu-Ni nanofluids.

**Figure 7 nanomaterials-12-02570-f007:**
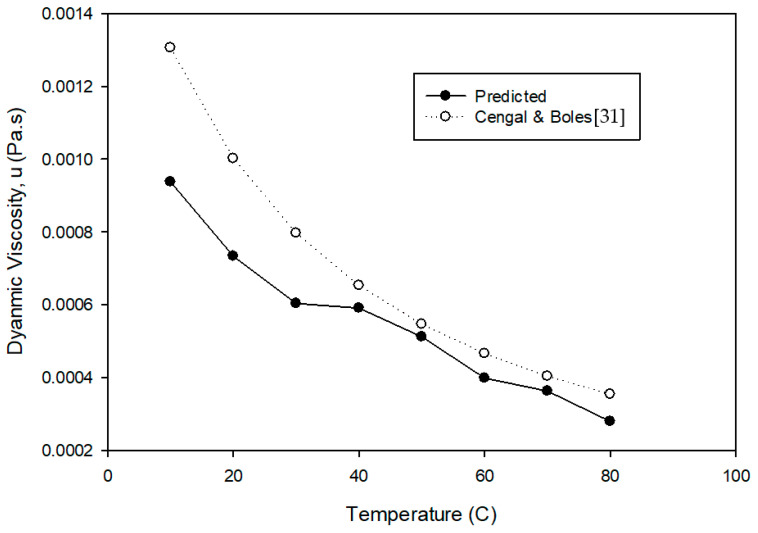
Predicted base fluid viscosity at different temperatures.

**Figure 8 nanomaterials-12-02570-f008:**
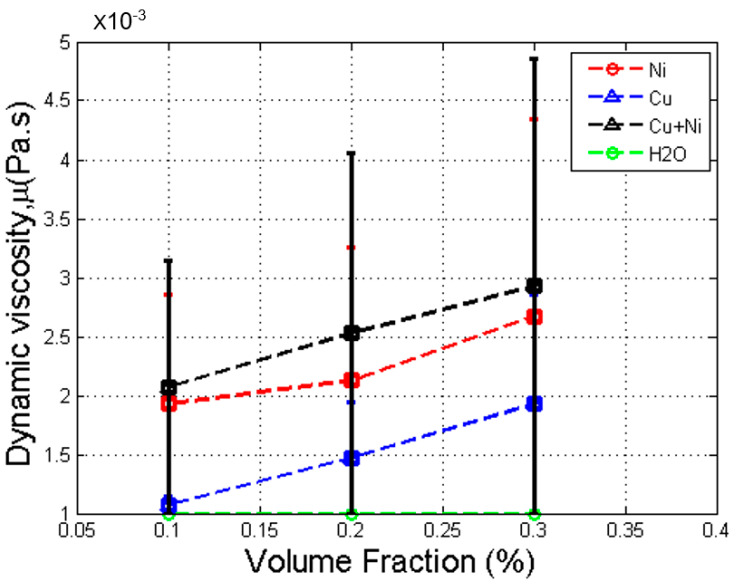
Dynamic viscosity versus volume fraction for Cu, Ni, and Cu-Ni nanofluids with error bars.

**Figure 9 nanomaterials-12-02570-f009:**
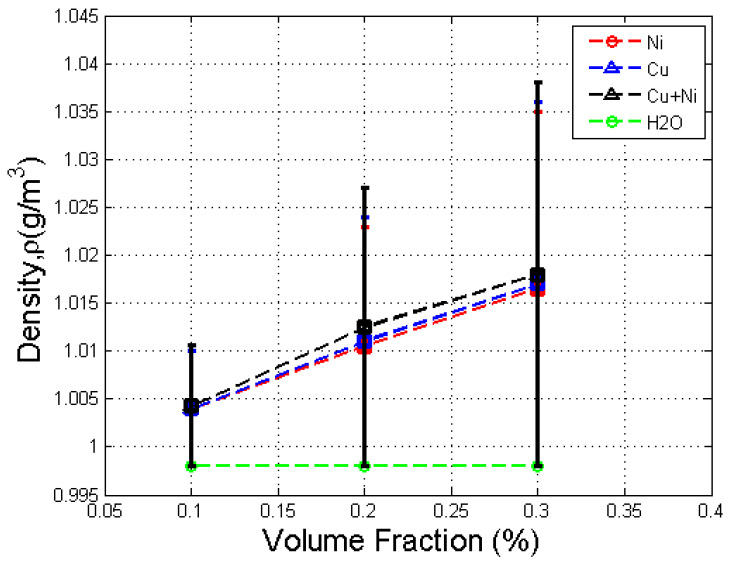
Density against volume fraction for Cu, Ni, and Cu-Ni nanofluids with error bar.

**Table 1 nanomaterials-12-02570-t001:** Molar masses for hydrogen, oxygen, copper, and nickel particles.

Type of Particle	Molar Mass (g/mol)
Hydrogen (H)	1.008
Oxygen (O)	15.9994
Copper (Cu)	63.55
Nickel (Ni)	58.6934

**Table 2 nanomaterials-12-02570-t002:** EAM potential files used for Cu-Cu and Ni-Ni interactions [28].

Interactions	Name of EAM Potential File
Cu-Cu	Cu_zhou.eam.alloy
Ni-Ni	NiAlH_jea.eam.alloy

**Table 3 nanomaterials-12-02570-t003:** LJ potential parameters for Cu-H_2_O, Ni-H_2_O and Cu-Ni cross-interactions in this work.

Cross-Interactions	Interaction Strength, ε (kcal/mol)	Interaction Length Scale, σ (A)
Cu-H_2_O	0.80	1.60
Ni-H_2_O	0.80	1.30
Cu-Ni	1.00	0.70

**Table 4 nanomaterials-12-02570-t004:** Predicted and actual density of water.

Fluid	$\mathrm{Predicted}\;\mathrm{Density},\;\rho_{s}\;(g/{\mathrm{cm}}^{3})$	$\mathrm{Actual}\;\mathrm{Density},\;\rho_{a}\;(g/{\mathrm{cm}}^{3})$ [28]
Water	0.99656	0.9982

## Data Availability

Not applicable.

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
