# Peer review of "Influence of Nanoparticles on Thermophysical Properties of Hybrid Nanofluids of Different Volume Fractions"

_nanomaterials, 2022, doi:10.3390/nano12152570_

Round 1
Reviewer 1 Report
1. The text must be revised for the English usage.
2. In title must be: “…of Different Volume Fractions” instead of “…for Different Volume Fraction”
3. It is unusual for Cu–Ni hybrid nanofluid, thermal conductivity increases with volume fraction and is larger than that of base fluid and their respective mono nanofluids (Figure 5, p. 7). The difference in data is less than 0.001%. The authors must show the error bars. Further studies are needed to explain this phenomenon.
4. The data provided in Figures 5 and 6 are not consistent for Cu+Ni nanofluid.
5. Markers for H2O and Cu must be different in Figure 8. It will be good to show the data for Ni in Figure 8. Error bars must be shown.
6. Error bars must be shown in Figure 9.
7.
Author Response
Reviewers' Comments and Authors’ Replies
Reviewer #1 comment no. 1: The text must be revised for the English usage.
Authors’ reply to comment no. 1: Thank you for reviewing and giving constructive comments that are helpful to improve the quality of the manuscript. Thank you for giving the opportunity to revise our manuscript. We have done our best to improve it. Please find below the answers to the comments, point by point. The amendments are highlighted in red color. We look forward to your feedback on the revised manuscript.
Reviewer #1 comment no. 2: In title must be: “…of Different Volume Fractions” instead of “…for Different Volume Fraction”
Authors’ reply to comment no. 2: Thank you for your constructive comment. The title is now amended to “Influence of Nanoparticles on Thermophysical Properties of Hybrid Nanofluids of Different Volume Fraction”.
Reviewer #1 comment no. 3: It is unusual for Cu–Ni hybrid nanofluid, thermal conductivity increases with volume fraction and is larger than that of base fluid and their respective mono nanofluids (Figure 5, p. 7). The difference in data is less than 0.001%. The authors must show the error bars. Further studies are needed to explain this phenomenon.
Authors’ reply to comment no. 3: Thank you for your constructive comment. The errors bars are added in Figure 5 (p8). The different in LJ potentials and atom’s radius (Van der Waals) between Cu and Ni are relatively small. These could be contributed small different in both mono nanofluids. Yes, in our future study, we will need to focus in this issue.
Reviewer #1 comment no. 4: The data provided in Figures 5 and 6 are not consistent for Cu+Ni nanofluid.
Authors’ reply to comment no. 4: Thank you for highlighting this issue. Actually, we made the typo errors in the graph’s legend (Figure 6, p9). The figure has been corrected.
Reviewer #1 comment no. 5: Markers for H2O and Cu must be different in Figure 8. It will be good to show the data for Ni in Figure 8. Error bars must be shown.
Authors’ reply to comment no. 5: Thank you for your constructive comment. The improvement and error bars are shown in Figure 8 (p10).
Reviewer #1 comment no. 6: Error bars must be shown in Figure 9.
Authors’ reply to comment no. 6: Thank you for your constructive comment. The error bars are added in Figure 9 (p 11).
Reviewer 2 Report
This manuscript comprehensively reviews Influence of Nanoparticles on Thermophysical Properties of Hybrid Nanofluids for Different Volume Fraction. Even if this topic is still quite incipient, a review paper could provide some light for future researchers in the topic. I have the following comments for authors:
1. Please add some more quantitative results in the introduction, it seems to be stacked.
2.Check full text for grammar.
3. In order to make the Introduction section more complete, the reviewers suggest citing the following article: https://doi.org/10.1016/j.aca.2022.339927.
4. It is best to explain the practical significance of this study.
Author Response
Reviewer #2: This manuscript comprehensively reviews Influence of Nanoparticles on Thermophysical Properties of Hybrid Nanofluids for Different Volume Fraction. Even if this topic is still quite incipient, a review paper could provide some light for future researchers in the topic. I have the following comments for authors:
Authors’ reply: Thank you for reviewing and giving constructive comments that are helpful to improve the quality of the manuscript. Thank you for giving the opportunity to revise our manuscript. We have done our best to improve it. Please find below the answers to the comments, point by point. The amendments are highlighted in red color. We look forward to your feedback on the revised manuscript.
Reviewer #2 comment no. 1: Please add some more quantitative results in the introduction, it seems to be stacked.
Authors’ reply to comment no. 1: Thank you for your constructive comment. The quantitative results in the introduction are added (marked in red).
Reviewer #2 comment no. 2: Check full text for grammar.
Authors’ reply to comment no. 2: Thank you for your constructive comment. We have attempted to reduce the grammatical errors to our best ability.
Reviewer #2 comment no. 3: In order to make the Introduction section more complete, the reviewers suggest citing the following article: https://doi.org/10.1016/j.aca.2022.339927.
Authors’ reply to comment no. 3: Thank you for your constructive comment. The suggested article is cited in the paper (lines 74-80).
Reviewer #2 comment no. 4: It is best to explain the practical significance of this study.
Authors’ reply to comment no. 4: Thank you for the comment. The practical significance of this study is highlighted in the introduction section as suggested (lines 87-90).
Round 2
Reviewer 1 Report
Paper is recommended for publication.